# How Do the Timing of Early Rehabilitation Together with Cognitive and Functional Variables Influence Stroke Recovery? Results from the CogniReMo Italian Multicentric Study

**DOI:** 10.3390/healthcare13030316

**Published:** 2025-02-04

**Authors:** Mauro Mancuso, Marco Iosa, Giovanni Morone, Daniela De Bartolo, Irene Ciancarelli

**Affiliations:** 1Physical and Rehabilitative Medicine Unit, South-East Tuscany Regional Health Service, 58100 Grosseto, Italy; m.mancuso62@gmail.com; 2Tuscany Rehabilitation Clinic, 52025 Montevarchi, Italy; 3Department of Psychology, Faculty of Medicine and Psychology, Sapienza University of Rome, 00185 Rome, Italy; 4Smart-Lab, Santa Lucia Foundation-IRCCS, 00179 Rome, Italy; 5Department of Life, Health and Environmental Sciences, University of L’Aquila, 67100 L’Aquila, Italy; giovanni.morone@univaq.it (G.M.); irene.ciancarelli@univaq.it (I.C.); 6San Raffaele Istitute of Sumona, 67039 Sulmona, Italy; 7Department of Developmental and Social Psychology, Faculty of Medicine and Psychology, Sapienza University of Rome, 00185 Rome, Italy

**Keywords:** rehabilitation, stroke, cognitive assessment, attention, disability, motor recovery, prognostic factors

## Abstract

**Background:** The time lapse between the acute event and the beginning of rehabilitation seems to play a significant role in determining the effectiveness of rehabilitation together with the severity of neurological deficits and impairments of motor and cognitive functions. The present study aims to further explore the prognostic role of cognitive and motor functions, concerning the different times of the beginning of neurorehabilitation. **Methods:** A secondary examination was conducted by applying a cluster analysis on the data of 386 stroke patients in the subacute phase who were enrolled in the Cognitive and Recovery of Motor Functions (CogniReMo) study. **Results:** The Barthel Index at the admission predicts clinical outcome: if BI was 0, it was on average 28.7 ± 24.1 at discharge. For patients with Barthel Index <15 at discharge, the discriminant was unaltered executive functions having an average output of 61.3 instead of 45.5. In the range of BI at admission between 16 and 45, the discriminant variable was to have an NIHSS ≤ 5 to obtain a high outcome (BI = 75.4 instead of BI = 61.9). Subjects with a BI at admission >45 were the best responders to rehabilitation, with a mean BI at discharge of 85 if they have alteration in spatial attention, and 95.3 if they have no deficits in spatial attention. Also, for inpatients hospitalized in a period ranging from the 20th to the 37th day after stroke, spatial attention was a discriminant variable to have a poor outcome (BI = 34.3) vs. a good one (BI = 76.7). **Conclusions:** The algorithm identified a hierarchical decision tree that might assume a significant role for clinicians in defining an appropriate rehabilitation pathway, depending on the time of rehabilitation beginning and the severity of motor and cognitive deficits.

## 1. Introduction

Stroke is the leading cause of adult disability in Europe and worldwide, resulting in a significant loss of independence in activities of daily living [1]. The consequences of stroke affect not only individuals but also their caregivers and families, leading to a significant increase in the socioeconomic burden [2].

Patients who experience severe stroke often face impairments in both motor and cognitive functions [3]. Among motor impairments, ambulatory functions are often recovered within a few weeks, even after severe hemiplegia, whereas the restoration of upper limb function remains highly challenging [3,4]. Cognitive functions, such as attention, memory, language, executive functions, and spatial awareness, are often impaired, significantly impacting the effectiveness of recovery and social participation [5]. Additionally, patients may often experience fatigue, difficulties in swallowing, behavioral disorders, and alterations in emotional functioning, including depression and anxiety [4]. These factors contribute to reduced subjective well-being, increased psychological distress, and greater dependence on caregivers for daily activities [6].

The severity of functional consequences after stroke still lacks substantial evidence, although certain topics have begun to emerge in recent years. Among the variables that should be considered, the severity of stroke seems to play a significant role, with notable effects on quality of life that extend beyond functional limitations and mortality outcomes [7]. Understanding the relationship between prognostic factors and clinical outcomes is essential for guiding rehabilitation, optimizing patient care, and ensuring the appropriate use of resources [3,8].

Scientific literature identifies several negative prognostic factors for rehabilitation outcomes, including the severity of neurological motor deficits, age, gender, BMI, premorbid conditions, concomitant pathologies [8,9], prolonged time between the acute event and the start of rehabilitation, global aphasia, the presence of spatial neglect, and depression [10].

Beyond these variables, cognitive impairment plays a significant role in determining both stroke severity and the effectiveness of rehabilitative interventions. Properly assessing cognitive functions serves as a significant predictor of overall recovery and social participation [11]. Moreover, impairments in memory, language, and praxis have been shown to influence recovery after stroke [12]. A lack of recovery in executive functions following rehabilitative interventions also adversely affects motor recovery [13]. Notably, the re-acquisition of motor skills relies on learning mechanisms that are strictly intertwined with cognitive functions, particularly with executive functions, which are fundamental in determining the success of rehabilitation, and also those focused on motor functions [5].

Rehabilitation is currently the most effective intervention for promoting recovery in stroke patients [14], and its effectiveness appears to be influenced by the time lapse between the acute event and the beginning of therapy. In fact, some authors have highlighted that the first three months following stroke onset represent the golden period for functional recovery, due to cortical plasticity [15]. The timeline of rehabilitation has generally been divided into three periods: very early rehabilitation, early rehabilitation, and late rehabilitation. Very early rehabilitation is defined as rehabilitative interventions delivered within 24 h after a stroke [16]. Early rehabilitation refers to interventions provided within the first two weeks after a stroke, whereas late rehabilitation is defined as rehabilitation delivered after the first two weeks following a stroke [15].

In this context, very early rehabilitation should be provided with caution, as it may be less effective and potentially inappropriate due to the phenomena associated with the penumbra area and the toxic effect of glutamate in the first 16 h following stroke onset [17]. Conversely, early mobilization conducted within stroke units and followed by expedited early rehabilitation in specialized rehabilitative wards is recommended by the majority of clinical practice guidelines. This approach is associated with improved recovery outcomes for patients after a stroke [15]. Early rehabilitation appears to be more effective than late rehabilitation because it is implemented during periods of heightened brain plasticity. Furthermore, early interventions can prevent or reduce secondary damages to the musculoskeletal, cardiovascular, respiratory, and immune systems caused by prolonged bed rest [18].

Lastly, neuroscientific theories regarding inter-hemispheric bimodal balance and its modification over time suggest a different behavior of inter-hemispheric connectivity in relation to the severity of cerebral lesions [19].

Despite the division into three phases, the time lapse between the acute event and the beginning of rehabilitation is a continuous variable. In clinical practice, it is often influenced more by administrative issues related to the management of the clinical pathway than by demographic or clinical factors. Hence, there is a need for scientific insight into the precise mechanisms underlying the association between the time elapsed since the acute event and other prognostic factors, such as severity and cognitive functions, which are essential for predicting outcomes but remain incompletely understood.

In this study, we aimed to further explore the prognostic roles of stroke severity, cognitive functions, and motor functions, with a specific focus on functional recovery. To achieve this, we analyzed a cohort of stroke patients stratified according to the timing of neurorehabilitation initiation. Decisional cluster analysis was employed, as it has demonstrated superior accuracy in identifying stroke prognostic factors compared to classical regression and neural network analyses [20].

## 2. Materials and Methods

### 2.1. Participants

This study reports a secondary analysis of data from a cohort of 386 stroke patients in the subacute phase (discharged from acute wards, i.e., stroke units or neurology wards), who were consecutively admitted from November 2019 to July 2021 to sixteen Italian rehabilitation centers. These patients were enrolled in the observational, prospective, longitudinal, multicentric study named Cognitive and Recovery of Motor Functions (CogniReMo). The cohort included both genders, aged between 18 to 90 years, with varying levels of education. Sixty-seven patients dropped out and were transferred to other hospitals due to clinical worsening (26 females and 41 males). The final study group consisted of 319 patients (129 females and 190 males).

Inclusion criteria: Patients with ischemic stroke aged between 18 and 90 years were enrolled. Both males and females of any educational background were admitted.

Exclusion criteria: Patients with hemorrhagic stroke, previous psychiatric or neurological disorders, a prior Barthel Index score lower than 90, or a history of drug or alcohol abuse were excluded.

Patients admitted to rehabilitative wards 72 h before or 90 days after stroke onset were excluded. Additionally, patients who did not provide informed consent were also excluded.

The study was approved by the Ethical Committee of South-East Tuscany under number 15360. All patients were informed about the aims of the study, and all participants signed the informed consent.

### 2.2. Procedure

The severity of stroke for each patient enrolled in the study was assessed using the National Institutes of Health Stroke Scale (NIHSS). The locations of brain lesions were classified according to the Bamford classification [21]. Motor impairment was evaluated with the Fugl–Meyer Motor Scale [22], while disability was assessed using the Barthel Index [23]. Cognitive impairments were screened with the Oxford Cognitive Screen (OCS) [24]. All the assessment scales were administered at both admission and discharge. We collected demographic information for each patient (age, gender, education level, previous Barthel Index scores) and risk factors such as hypertension, diabetes, obesity (body mass index: BMI), and atrial fibrillation. Clinical information like stroke onset, admission, and discharge dates was also collected, along with information regarding acute phase treatments (fibrinolysis and/or thrombolysis), clinical complications (such as deep vein thrombosis, pulmonary embolism, bladder infection, pulmonary infection, and septicemia), as well as discharge settings.

All the evaluation scales were administered and scored by members of the patient’s rehabilitation care team who were blinded to the aim of the study. Additionally, the therapists were also blinded to the study’s aim.

The baseline assessment (T0) and follow-up evaluation were performed within 48 h of admission and before discharge, respectively.

During the hospital recovery, all patients underwent conventional multidisciplinary rehabilitation treatment. According to the Italian guidelines, conventional treatment for acute stroke patients foresees a daily treatment of 3 h per day, 6 days per week. Physiotherapy, when it is possible and safe for the patient’s care, usually starts within 24 h of admission, while training for cognitive rehabilitation, as well as training for swallowing, bowel, and bladder function recovery, is included at a later stage. Additionally, occupational therapy is provided for all patients. For patients enrolled in this study, physiotherapy training exercises for improving trunk control, standing, balance, gait, coordination, and endurance were provided. Patients completed cognitive training aimed to improve cognition, attention, memory, spatial awareness, language, praxis, and executive functions. Finally, swallowing treatment was performed for approximately 30 min per day, when necessary, along with speech therapy for patients affected by aphasia as well as specific psychological interventions for those with unilateral spatial neglect.

### 2.3. Measures

We performed a clinical assessment of enrolled patients using the National Institutes of Health Stroke Scale (NIHSS) [25], for evaluation of stroke severity. This scale allows the rapid screening of patients, based on 11 items. Scores near 0 stand for normal neurological functioning, while those close to 42 indicate severe neurological damage.

For the assessment of cognitive function, we administered the Oxford Cognitive Screen (OCS) [24] Italian version [26] including a normative sample of individuals aged 20 to 80 years. This scale assesses cognitive functioning based on five domains: attention and executive function, language, memory, number processing, and praxis. No scaling corrections are applied to the raw data. This scale is easy to use in the acute phase following a stroke event, as it takes only 15 min of administration and can be administered even at the patient’s bedside whenever necessary.

For quantification of motor impairment, we administered the Fugl–Meyer Assessment (FMA) Italian version [27]. This tool provides a quantitative examination of the residual motor ability assessed through the direct observation of motor performance in specific tasks involving precise muscles and synergies with regard to the upper and lower limb function. The Fugl–Meyer Assessment for the Lower Extremity (FMA-LE) has a maximum score of 34 points, which is based on lower extremity scores (0–28 points) and coordination/speed scores (0–6 points). The non-motor domains of the Fugl–Meyer Assessment include Sensation and Joint Passive. Joint Movement and Pain are assessed for both the upper and lower limbs.

The modified Barthel Index (mBI) [27] is an ordinal scale used to measure performance and independence in activities of daily living (ADLs). The BI scores ten variables that describe ADL and mobility. A higher score corresponds to greater independence in performing ADL. Scores range from 0 to 100 based on the level of assistance a patient requires in performing the tasks. Higher scores indicate greater autonomy in conducting daily activities.

The reliability of the clinical assessments corresponded to the psychometric properties of the administered scales previously evaluated in the validation studies for each scale and test [22,23,24,25,26,27].

## 3. Statistical Analysis

Data are reported as mean ± standard deviation or percentage relative frequencies. Pre- vs. post-rehabilitation scores were compared using a non-parametric Wilcoxon test. The alpha level of significance was set at 5% for all the analyses. A cluster analysis was performed utilizing a decisional tree based on the exhaustive CHAID growing model. The minimum number of subjects allowed for parent and child nodes was 20 (6% of the whole sample). The output variable was the Barthel Index score at discharge; according to the aim of this study, the time from stroke at admission was set as the primary decisional variable, whereas 18 other variables recorded at admission have been further included as possible decisional variables such as Age, Gender, Barthel Index, NIHSS, Lower Limb Fugl–Meyer, and the OCS domains: Picture Naming, Semantics, Orientation, Visual Field, Spatial Attention (broken heart cancellation test score and time), Reading, Number Writing, Calculation, Praxis, Verbal Memory, Episodic Memory, Executive Functions. For each cluster, we computed the effect size of rehabilitation (difference between BI scores at discharge and admission divided by the standard deviation at admission) and the effectiveness (the percentage difference between BI scores at discharge and admission divided by the maximum achievable score minus the admission score).

## 4. Results

Table 1 summarizes the demographic and clinical characteristics of the patients at the time of admission to rehabilitation hospitals. At discharge, the mean Barthel Index was 64.6 ± 29.7, showing a significant improvement compared to the score assessed at admission (*p* < 0.001).

Figure 1 shows the results of the cluster analysis, which identified four main time windows. The first cluster was associated with early rehabilitation, which led to a high outcome (BI = 82.9 ± 24.1). The last cluster is associated with a delayed start of rehabilitation (approximately 10% of subjects admitted to the rehabilitation hospital after more than 37 days post-stroke), resulting in a low outcome (BI = 43.2 ± 28.7).

Most patients (60%) were admitted to the rehabilitation hospital between 7 and 19 days post-stroke. In these patients, the Barthel Index (BI) score at admission was the most significant predictor of their outcomes, as reported in Table 2. Specifically, if the BI was 0 at admission, it increased to an average of 28.7 ± 24.1 at discharge. For patients with a Barthel Index (BI) <15 at admission, the discriminant factor for achieving an average outcome was to have unaltered executive functions, resulting in an average BI of 61.3 compared to 45.5 of patients with a deficit in executive functions. For patients with a BI at admission between 16 and 45, having an NIHSS ≤5 was the key discriminant variable for obtaining a high outcome, with an average BI of 75.4 instead of 61.9. Subjects with a BI at admission >45 were the best responders to therapy, achieving a mean BI at discharge of 85 if they had alterations in spatial attention, and 95.3 if they had no deficits in spatial attention.

The remaining patients (21%) were admitted to the rehabilitation hospital between the 20th and 37th days after the acute event. For these patients, spatial attention was a discriminant variable influencing outcome; those with spatial attention deficits had a poor outcome, with an average Barthel Index (BI) of 34.3, while those without deficits achieved a better outcome, with an average BI of 76.7.

## 5. Discussion

The aim of this study was to investigate how the timing of rehabilitation onset influences the impact of established prognostic factors, such as the severity of neurological deficits and cognitive impairments, on rehabilitative effectiveness.

Our cluster analysis identified four critical time windows within the framework of early rehabilitation. These windows revealed distinct factors that could be considered prognostic indicators of outcomes. Patients who initiated rehabilitation within one week of the acute event were good responders, regardless of other influencing factors. Conversely, patients who began rehabilitation later than 37 days after the acute event (approximately 5.5 weeks) were identified as low responders, regardless of other influencing factors. These findings are in accordance with those found by Chen et al. [10]. Most patients began rehabilitation between 7 and 19 days after the acute event, and their outcomes were influenced by their level of autonomy (measured by the Barthel Index), stroke severity (assessed using the NIHSS), and their executive functions and spatial attention at the time of admission to the rehabilitation hospital. Spatial attention was also a significant factor for patients who started therapy between 20 and 37 days post-stroke. So, four out of the eighteen variables have been identified as discriminants for the outcome. In a previous study [13], we identified five variables based on linear regression analysis, three of which were also identified by the present cluster analysis: the BI at admission, the OCS score for executive functions, and the OCS score for the broken heart cancellation task. The other two variables previously identified were the time taken to complete the OCS broken heart cancellation task (related to spatial attention) and the Lower Limb Fugl–Meyer score. The latter did not enter the model obtained by the cluster analysis, likely because it was replaced by the NIHSS score.

It is noteworthy that the cluster analysis automatically identified some interesting thresholds for these other variables that are aligned with those reported in the scientific literature.

The NIHSS is conventionally divided into minor (0–4 score) and moderate/severe strokes (>4 score). The cluster analysis identified a threshold of 5, which is very close to the conventionally used classification.

Mancuso et al. (2016) [26] reported that the Executive Functions were altered when the corresponding OCS score was or ≤−4 or ≥4. In our analysis, we used the absolute values, and the identified threshold was exactly 4.

For Spatial Attention, for the score of the heart cancellation task, Mancuso et al. (2016) [26] reported the following equation: score = 46.925 − 0.03 × Age + 0.147 × Education. Considering the mean age and education of our sample (71 years old and 9 years of school) we obtain a cut-off level of 46, which is exactly the threshold found for patients with a BI at admission (>45) for differentiating between high responders (BI at discharge = 85) and very high responders (95.3). The other threshold identified by the cluster analysis for spatial attention was 24, which is a low value likely indicating a severe deficit in spatial attention, as 57.7% of these patients exhibited neglect.

The most important finding of this study is that the prognostic variables of post-stroke functional recovery do not carry a fixed weight for all patients; rather, their impact depends on the timing of the initiation of the rehabilitation intervention. This finding is significant as it adds a novel perspective to the discussion surrounding early rehabilitation and, more broadly, the timing of rehabilitation in stroke care to optimize recovery.

This result deserves to be underlined, especially considering that the time from the acute event to the beginning of rehabilitation may depend not only on clinical factors, such as the stabilization of the patients’ critical conditions, but also on local administrative issues related to hospitalization procedures in rehabilitation settings.

In fact, we should consider the specific prognostic factors for each time window in which rehabilitation is initiated. This principle aligns with an evaluation of predictive algorithms that adhere to the principles of personalized medicine. Our algorithm is an effective and easily usable tool in daily clinical practice. Unlike the Predicting REcovery Potential PREP model [28], which primarily focuses on predicting outcomes related to the upper limb, our algorithm predicts overall patient outcomes. It encompasses key cognitive aspects relevant to activities of daily living (ADLs) and quality of life. Furthermore, while the PREP model includes neurophysiological factors that may not always be readily available, our algorithm does not incorporate these variables, making it more accessible across diverse clinical settings.

Our study also highlighted the importance of prompt and accurate assessment of cognitive functions shortly after a stroke and their effective rehabilitation. Specifically, in patients managed between 7 and 35 days after the event, our algorithm demonstrated that early management (7–19 days) is primarily influenced by the Barthel Index at entry—if it is 0. For patients with a Barthel Index between 1 and 15, outcomes are influenced by executive functions; for those with a Barthel Index between 16 and 45, the severity of the neurological deficit as measured by the NIHSS plays a critical role. Additionally, for patients with a Barthel Index greater than 45, spatial neglect becomes a significant factor.

In patients managed between 20 and 37 days post-stroke, the presence of spatial function deficits is decisive for outcomes. Conversely, both very early and late management are less influenced by the interplay between the timing of onset and cognitive functions. In summary, cognitive functions directly modify the effectiveness of early rehabilitation when patients begin rehabilitation between 20 and 37 days after the stroke. For those starting rehabilitation between 7 and 19 days, cognitive functions play a subordinate role, primarily affecting outcomes based on the Barthel Index.

Our model did not consider age as a significant prognostic factor. Our inclusion criteria allowed the enrollment of patients aged >18 years; however, the youngest enrolled patient was 39 years old, and the sample was relatively homogeneous in terms of age (70.9 ± 11.0 years). It may have limited the effect of age on the outcome. Future studies on larger samples could be conducted by grouping patients by age for more detailed cluster analyses.

The findings of this study should be interpreted with caution in light of its limitations. First, the database used to identify the clusters was not prospectively validated with other datasets, which may affect the accuracy of the proposed model. Second, despite the considerable sample size, some clusters were derived from a limited number of subjects, potentially impacting the robustness of our findings. Furthermore, certain clinical variables, such as lesion volume, were not accounted for in our analyses. Lastly, although the study was multicentric, all participating hospitals were located in Italy. This means that the clinical management of patients adhered to Italian laws and guidelines, which may restrict the generalizability of the findings to other countries.

## 6. Conclusions

The algorithm identified in this study has the potential to support clinicians in determining optimal rehabilitation pathways by considering the timing of rehabilitation initiation and the severity of motor and cognitive impairments.

Additionally, it may serve as a valuable source of information for policymakers in establishing clinical practice guidelines, emphasizing the necessity of resource allocation and bed availability to enable the timely initiation of rehabilitation within a few days following the acute event.

## Figures and Tables

**Figure 1 healthcare-13-00316-f001:**
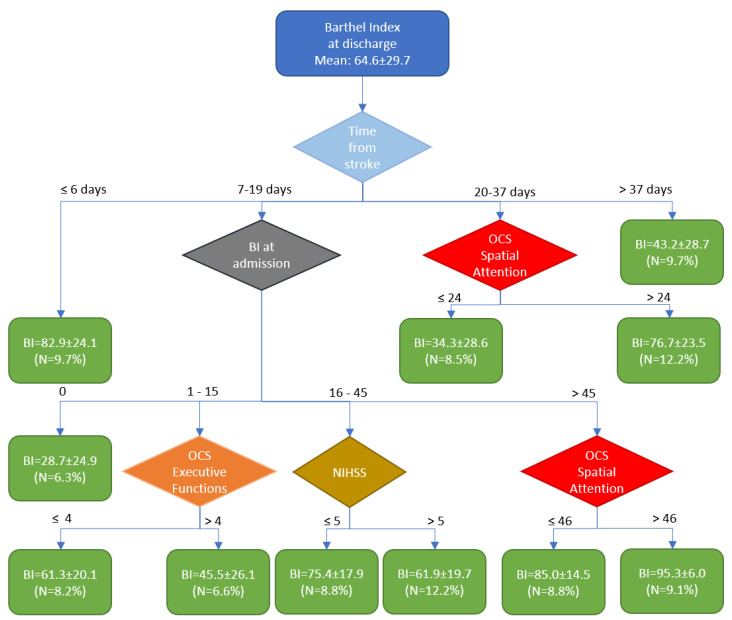
The figure shows the tree obtained by cluster analysis (BI: Barthel Index, NIHSS: National Institute of Health Stroke Scale, OCS: Oxford Cognitive Screen, N: the percentage number of subjects included in the cluster; the green rectangles are the final clusters, and the rhomboids correspond to decisional nodes).

**Table 1 healthcare-13-00316-t001:** Demographic and clinical variables at the admission into the rehabilitation hospital. Abbreviations in the table are TACIs (total anterior circulation infarcts), LACIs (lacunar infarcts), PACIs (partial anterior circulation infarcts), POCIs (posterior circulation infarcts), NIHSS (National Institutes of Health Stroke Scale).

Demographic and Clinical Variables	
Age	70.9 ± 11.0 years
Schooling time	9.0 ± 4.3 years
Gender	44% of females
Time from stroke and rehabilitation	18.6 ± 15.8 days
Length of stay in the hospital	49.6 ± 32.6 days
Side of stroke	50.3% right, 44.6% left, 5.1% bilateral
Bamford Classification	TACIs: 20.3%, LACIs: 22.1%, PACIs: 40.2%, POCIs: 17.4%
Clinical conditions	84% hypertension; 21.9% diabetes 21.0% atrial fibrillation
Acute intervention Barthel Index NIHSS	19.9% thrombectomy; 25.5% fibrinolysis
	33.2 ± 28.0
	7.9 ± 5.5

**Table 2 healthcare-13-00316-t002:** Effect size and effectiveness of the clusters identified by timing of rehabilitation onset since stroke acute event, Barthel Index (BI) score, score of OCS Executive Functions (OCS Ex Funct), score of OCS Spatial Attention (OCS Sp Att), NIH Stroke Scale (NIHSS) score at admission.

Timing of Rehabilitation Onset	Functional Variables at Admission	Cognitive Variables at Admission	Effect Size	Effectiveness
≤6 days	-	-	0.914	62.7%
7–19 days	BI = 0	-	-	28.7%
	BI = 1–15	OCS Ex Funct ≤ 4	13.696	56.9%
	BI = 1–15	OCS Ex Funct > 4	15.316	43.8%
	BI = 16–45	NIHSS ≤ 5	4.544	63.0%
	BI = 16–45	NIHSS > 5	4.644	45.3%
	BI > 45	OCS Sp Att ≤ 46	0.801	49.0%
	BI > 45	OCS Sp Att >46	2.055	83.9%
20–37 days	-	OCS Sp Att ≤ 24	1.819	25.1%
20–37 days	-	OCS Sp Att >24	1.401	61.9%
>37 days	-	-	1.590	32.4%

## Data Availability

Data will be available upon reasonable request made to the corresponding authors.

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
