# Peer review of "How Do the Timing of Early Rehabilitation Together with Cognitive and Functional Variables Influence Stroke Recovery? Results from the CogniReMo Italian Multicentric Study"

_healthcare, 2025, doi:10.3390/healthcare13030316_

Round 1

Reviewer 1 Report

Comments and Suggestions for Authors

Dear Authors,

The paper investigates stroke rehabilitation with respect to the prognostic roles of stroke severity, cognitive functions, and motor functions, with a specific focus on functional recovery.

Although the manuscript has a value for healthcare professionals, it needs some improvements.

Readers will appreciate if authors add a mandatory section "Conclusions".

Also, providing limitations of the study might strengthen the paper.

Regards,

Reviewer

Author Response

Dear Authors,

The paper investigates stroke rehabilitation with respect to the prognostic roles of stroke severity, cognitive functions, and motor functions, with a specific focus on functional recovery.

 Although the manuscript has a value for healthcare professionals, it needs some improvements.

AUTHORS: We would like to thank the reviewer for the positive general comment about our manuscript and for the qualified suggestions that helped us to improve our work in this revised version. In the following our point by point responses (in blue) to the comments of the Reviewer (in black) in which we have also reported the new/changed parts of the revised manuscript (between apices and in italic style)

 Readers will appreciate if authors add a mandatory section "Conclusions".

AUTHORS: Thanks for this suggestion. We have added this section:

5. Conclusions

The algorithm identified in this study has the potential to support  clinicians in determining optimal  rehabilitation pathways by considering  the timing of rehabilitation initiation and the severity of motor and cognitive impairments.

Additionally, it may serve as a valuable resource for policymakers in establishing clinical practice guidelines, emphasizing the necessity of resource allocation and bed availability to enable the timely initiation of rehabilitation within a few days following the acute event.”

Also, providing limitations of the study might strengthen the paper.

AUTHORS: According to this comment we have now strengthened the limitations of our study as follows:
“The findings of this study should be interpreted with caution in light of  its limitations. First, the database used to identify the clusters was not prospectively validated with other datasets, which may affect the accuracy of the proposed model. Second, despite the considerable sample size, some clusters were derived from a limited number of subjects potentially impacting the robustness of our findings. Furthermore, certain clinical variables, such as lesion volume, were not accounted for in our analyses. Lastly, although the study was multicentric, all participating hospitals were located in Italy. This means that clinical management of patients adhered to Italian laws and guidelines, which may restrict the generalizability of the findings to other countries.”

Reviewer 2 Report

Comments and Suggestions for Authors

The authors of the article titled “ Can Cognitive and Functional Variables Influence Early Rehabilitation in Stroke Recovery? Results from the CogniReMo Italian Multicentric study” present the results of a secondary analysis of the data of a total of 386 stroke patients enrolled in from CogniReMo study database. After cluster analysis of available data researchers state, that post-stroke functional recovery depends predominantly on the timing of the initiation of the rehabilitation intervention. 

The article is interesting and addresses the extremely important subject of post-stroke rehabilitation and its timing. The study is multicentric and has a representative study group. However, I need to stress three elements that in my opinion need improvement to be considered for publication. 

First. The title is somehow misleading. In my opinion, authors should in some way include here the main finding - the timing of the application of rehabilitative treatment.  

Second. Although the general structure of the work is proper there is quite a bit of imbalance regarding the length of the introduction. Authors should consider rewriting, making it more brief. Some of the sections seem more appropriate for the discussion section. 

Finally, the language and style of writing are not at all times comprehensive. It could use some English editing. 

I would be happy to reassess the article after corrections to the issues mentioned above are made. 

Comments on the Quality of English Language

Needs editing and corrections

Author Response

The authors of the article titled “ Can Cognitive and Functional Variables Influence Early Rehabilitation in Stroke Recovery? Results from the CogniReMo Italian Multicentric study” present the results of a secondary analysis of the data of a total of 386 stroke patients enrolled in from CogniReMo study database. After cluster analysis of available data researchers state, that post-stroke functional recovery depends predominantly on the timing of the initiation of the rehabilitation intervention. 

 The article is interesting and addresses the extremely important subject of post-stroke rehabilitation and its timing. The study is multicentric and has a representative study group. However, I need to stress three elements that in my opinion need improvement to be considered for publication. 

AUTHORS: We would like to thank the reviewer for the positive general comment about our manuscript and for the qualified suggestions that helped us to improve our work in this revised version. In the following our point by point responses (in blue) to the comments of the Reviewer (in black) in which we have also reported the new/changed parts of the revised manuscript (between apices and in italic style)

First. The title is somehow misleading. In my opinion, authors should in some way include here the main finding - the timing of the application of rehabilitative treatment.  

AUTHORS: Thanks for this suggestion. According to this comment we have modified the title to highlight the importance of timing of rehabilitation. The revised title is: “How Do the Timing of Early Rehabilitation, Cognitive and Functional Variables Influence Stroke Recovery? Results from the CogniReMo Italian Multicentric study”

Second. Although the general structure of the work is proper there is quite a bit of imbalance regarding the length of the introduction. Authors should consider rewriting, making it more brief. Some of the sections seem more appropriate for the discussion section. 

AUTHORS: According to this suggestion we deleted from introduction a couple of redundant sentences (because information was reported also elsewhere), in particular the following ones:

“Nevertheless, among the factors useful in determining stroke severity, the most long-established prognostic indicator is the severity of neurological motor deficits.”

“Beyond the severity of neurological motor deficits, other clinical factors have emerged in influencing rehabilitative recovery in recent years, such as age, gender, BMI, premorbid conditions, concomitant pathologies etc. [8,9]”

“Moreover, the variables that appear to correlate with functional outcomes are often considered as independent from one another, rather than as interconnected components of a complex algorithm.”

Finally, the language and style of writing are not at all times comprehensive. It could use some English editing. 

I would be happy to reassess the article after corrections to the issues mentioned above are made. 

AUTHORS: An English Editing has been performed throughout the whole manuscript.

Reviewer 3 Report

Comments and Suggestions for Authors

- you should add conclusion to the article 

- add more information in introduction 

- add a table describing the sampling 

- use recent references if possible 

- adjust figures 

- conclusion must be detailed to the work done 

- use more demonstration and figures 

Author Response

AUTHORS: We would like to thank the reviewer for the qualified suggestions that helped us to improve our work in this revised version. In the following our point by point responses (in blue) to the comments of the Reviewer (in black) in which we have also reported the new/changed parts of the revised manuscript (between apices and in italic style)

- you should add conclusion to the article 

AUTHORS: Thanks for this suggestion. We have added the section Conclusions to the revised version of our manuscript:

5. Conclusions

The algorithm identified in this study has the potential to support  clinicians in determining optimal  rehabilitation pathways by considering  the timing of rehabilitation initiation and the severity of motor and cognitive impairments.

Additionally, it may serve as a valuable resource for policymakers in establishing clinical practice guidelines, emphasizing the necessity of resource allocation and bed availability to enable the timely initiation of rehabilitation within a few days following the acute event.”

- add more information in introduction 

AUTHORS: According to this suggestion we have added more information in Introduction especially about the role of cognitive functions and their potential impairment, reported in the following paragraph:

“Beyond these variables, cognitive impairment plays a significant role in determining both stroke severity and the effectiveness of rehabilitative interventions. Properly assessing cognitive functions serves as a significant predictor of overall recovery and social participation [11]. Moreover, impairments in memory, language, and praxis have been shown to influence recovery after stroke [12]. A lack of recovery in executive functions following rehabilitative interventions also adversely affects motor recovery [13]. Notably, the re-acquisition of motor skills relies on learning mechanisms that are strictly intertwined with cognitive functions, particularly with executive functions, which are fundamental in determining the success of rehabilitation, also those focus on motor functions [5].”

- add a table describing the sampling 

AUTHORS: Thanks for this suggestion. In the new version of the manuscript Table 1 and Table 2 are used to describe the sample at admission and in terms of effect of rehabilitation at discharge.

- use recent references if possible 

AUTHORS: We have updated the references using more recent ones. In particular we have now cited:

Lang, C. E., Waddell, K. J., Barth, J., Holleran, C. L., Strube, M. J., & Bland, M. D. (2021). Upper limb performance in daily life approaches plateau around three to six weeks post-stroke. Neurorehabilitation and neural repair, 35(10), 903-914

Chen, W. C., Hsiao, M. Y., & Wang, T. G. Prognostic factors of functional outcome in post-acute stroke in the rehabilitation unit. J Formos Med Ass 2022, 121(3), 670-678.

Brancaccio, A., Tabarelli, D., & Belardinelli, P. A new framework to interpret individual inter-hemispheric compensatory communication after stroke. J Pers Med 2022, 12(1), 59.

Instead of:

Feys, H.; De Weerdt, W.; Nuyens, G.; van de Winckel, A.; Selz, B.; Kiekens, C. Predicting Motor Recovery of the Upper Limb after Stroke Rehabilitation: Value of a Clinical Examination. Physiother Res Int 2000, 5, 1–18.

Paolucci, S., Antonucci, G., Pratesi, L., Traballesi, M., Lubich, S., & Grasso, M. G. Functional outcome in stroke inpatient rehabilitation: predicting no, low and high response patients. Cerebrovasc. Dis 1998, 8(4), 228-234.

Di Pino, G., Pellegrino, G., Assenza, G., Capone, F., Ferreri, F., Formi-ca, D., ... & Di Lazzaro, V. Modulation of brain plasticity in stroke: a novel model for neurorehabilitation. Nat Rev Neurol 2014, 10(10), 597-608.

- adjust figures 

AUTHORS: In the revised version of the manuscript, Figure 1 represents the results of the Cluster Analysis.

- conclusion must be detailed to the work done 

AUTHORS: As reported above we have added a specific section about it, also taking into account this comment. The new section is the following:

5. Conclusions

The algorithm identified in this study has the potential to support  clinicians in determining optimal  rehabilitation pathways by considering  the timing of rehabilitation initiation and the severity of motor and cognitive impairments.

Additionally, it may serve as a valuable resource for policymakers in establishing clinical practice guidelines, emphasizing the necessity of resource allocation and bed availability to enable the timely initiation of rehabilitation within a few days following the acute event.”

- use more demonstration and figures 

AUTHORS: We have better detailed the demonstration reporting the following Table 2 that is strictly related to the groups reported in Figure 1.

Table 2. Effect size and Effectiveness of the clusters identified by timing of rehabilitation onset since stroke acute event, Barthel Index (BI) score, score of OCS Executive Functions (OCS Ex Funct), score of OCS Spatial Attention (OCS Sp Att), NIH-Stroke Severity score (NIHSS) at admission.

Timing of Rehabilitation onset

Functional Variables at admission

Cognitive Variables at admission

Effect Size

Effectiveness

≤ 6 days

-

-

0.914

62.7%

7-19 days

BI = 0

-

-

28.7%

BI = 1-15

OCS Ex Funct ≤ 4

13.696

56.9%

BI = 1-15

OCS Ex Funct > 4

15.316

43.8%

BI = 16-45

NIHSS ≤ 5

4.544

63.0%

BI = 16-45

NIHSS > 5

4.644

45.3%

BI > 45

OCS Sp Att ≤ 46

0.801

49.0%

BI > 45

OCS Sp Att >46

2.055

83.9%

20-37 days

-

OCS Sp Att ≤ 24

1.819

25.1%

20-37 days

-

OCS Sp Att >24

1.401

61.9%

> 37 days

-

-

1.590

32.4%

In the statistical analysis section we have reported the calculation of effect size:

“For each cluster, we computed the effect size of rehabilitation (difference between BI-scores at discharge and admission divided by the standard deviation at admission) and the effectiveness (the percentage difference between BI-scores at discharge and admission divided by the maximum achievable score minus the admission score).”

Reviewer 4 Report

Comments and Suggestions for Authors

This study is about whether cognitive and functional variables influence the rehabilitation of stroke patients. Cognition plays the most important role in the treatment of stroke patients. By confirming the influence of cognitive scores of early stroke patients on rehabilitation, we can convey important information to many clinicians.

Please consider the following:

1. If possible, please provide references for all sentences of the introduction you provided.

2. Please provide the role of cognition in the rehabilitation outcome of chronic stroke patients in the introduction.

3. If possible, please provide the effect size for the 386 subjects you provided.

4. Was the physiotherapy blinded to the study? Also, please provide the person who evaluated it.

5. It seems that the reliability of the evaluation items is needed.

6. Do you have MMSE data that can be included in Table 1?

7. This study targeted people between 18 and 90 years of age. It seems that the sample size was set large to gather a large number of people. However, it seems that the younger the person, the better the cognitive ability. Have you thought about this issue? It would be a good idea to conduct future research by grouping people by age.

Author Response

This study is about whether cognitive and functional variables influence the rehabilitation of stroke patients. Cognition plays the most important role in the treatment of stroke patients. By confirming the influence of cognitive scores of early stroke patients on rehabilitation, we can convey important information to many clinicians.

AUTHORS: We would like to thank the reviewer for the positive general comment about our manuscript and for the qualified suggestions that helped us to improve our work in this revised version. In the following our point by point responses (in blue) to the comments of the Reviewer (in black) in which we have also reported the new/changed parts of the revised manuscript (between apices and in italic style)

Please consider the following:

  1. If possible, please provide references for all sentences of the introduction you provided.

AUTHORS: Thanks for this suggestion, we have added some recent references related to the sentences of our manuscript, especially in Introduction. In particular, in the previous version of our manuscript we sometimes reported the reference at the end of a whole paragraph, we have now specified better the references related to each sentence.

  1. Please provide the role of cognition in the rehabilitation outcome of chronic stroke patients in the introduction.

AUTHORS: We have better specified this aspect in the following paragraph of Introduction of the revised manuscript:

“Beyond these variables, cognitive impairment plays a significant role in determining both stroke severity and the effectiveness of rehabilitative interventions. Properly assessing cognitive functions serves as a significant predictor of overall recovery and social participation [11]. Moreover, impairments in memory, language, and praxis have been shown to influence recovery after stroke [12]. A lack of recovery in executive functions following rehabilitative interventions also adversely affects motor recovery [13]. Notably, the re-acquisition of motor skills relies on learning mechanisms that are strictly intertwined with cognitive functions, particularly with executive functions, which are fundamental in determining the success of rehabilitation, also those focus on motor functions [5].”

  1. If possible, please provide the effect size for the 386 subjects you provided.

AUTHORS: We have added the Table 2 in which we have reported for each cluster the effect size and the effectiveness

Table 2. Effect size and Effectiveness of the clusters identified by timing of rehabilitation onset since stroke acute event, Barthel Index (BI) score, score of OCS Executive Functions (OCS Ex Funct), score of OCS Spatial Attention (OCS Sp Att), NIH-Stroke Severity score (NIHSS) at admission.

Timing of Rehabilitation onset

Functional Variables at admission

Cognitive Variables at admission

Effect Size

Effectiveness

≤ 6 days

-

-

0.914

62.7%

7-19 days

BI = 0

-

-

28.7%

BI = 1-15

OCS Ex Funct ≤ 4

13.696

56.9%

BI = 1-15

OCS Ex Funct > 4

15.316

43.8%

BI = 16-45

NIHSS ≤ 5

4.544

63.0%

BI = 16-45

NIHSS > 5

4.644

45.3%

BI > 45

OCS Sp Att ≤ 46

0.801

49.0%

BI > 45

OCS Sp Att >46

2.055

83.9%

20-37 days

-

OCS Sp Att ≤ 24

1.819

25.1%

20-37 days

-

OCS Sp Att >24

1.401

61.9%

> 37 days

-

-

1.590

32.4%

In the statistical analysis section we have reported the calculation of effect size:

“For each cluster, we computed the effect size of rehabilitation (difference between BI-scores at discharge and admission divided by the standard deviation at admission) and the effectiveness (the percentage difference between BI-scores at discharge and admission divided by the maximum achievable score minus the admission score).”

  1. Was the physiotherapy blinded to the study? Also, please provide the person who evaluated it.

AUTHORS: We have specified these aspects adding the following sentence:

“All the evaluation scales were administered and scored by members of the patient’s rehabilitation care team who were blinded to the aim of the study. Additionally, the therapists were also blinded to the study’s aim.”

  1. It seems that the reliability of the evaluation items is needed.

AUTHORS: We have added the following sentence about reliability of the evaluation tools:

“The reliability of the clinical assessments corresponded to the psychometric properties of the administered scales previously evaluated in the validation studies for each scale and test [22-27].”

  1. Do you have MMSE data that can be included in Table 1?

AUTHORS: Unfortunately, we did not assess the MMSE.

  1. This study targeted people between 18 and 90 years of age. It seems that the sample size was set large to gather a large number of people. However, it seems that the younger the person, the better the cognitive ability. Have you thought about this issue? It would be a good idea to conduct future research by grouping people by age.

AUTHORS: This is a very important point and we have added the following paragraph in the revised version of Discussion:

“Our model did not consider age as a significant prognostic factor. Our inclusion criteria allowed the enrolment of patients aged >18 years; however, the youngest enrolled patient was 39 years old, and the sample was relatively homogenous in terms of age (70.9±11.0 years). It may have limited the effect of age on the outcome. Future studies on wider samples could be conducted grouping patients by age for more de-tailed cluster analyses.”